# Neural Lad: A Neural Latent Dynamics Framework for Times Series Modeling

**Ting Li**
Ant Group, Beijing, China
`lt317068@antgroup.com`

**Jianguo Li**
Ant Group, Beijing, China
`lijg.zero@antgroup.com`

**Zhanxing Zhu**[✉]
Changping National Lab & Peking University, Beijing, China
`zhanxing.zhu@pku.edu.cn`

## Abstract

Neural ordinary differential equation (Neural ODE) is an elegant yet powerful framework to learn the temporal dynamics for time series modeling. However, we observe that existing Neural ODE forecasting models suffer from two disadvantages: i) controlling the latent states only through the linear transformation over the local change of the observed signals may be inadequate; ii) lacking the ability to capture the inherent periodical property in time series forecasting tasks; To overcome the two issues, we introduce a new neural ODE framework called **Neural Lad**, a **Neural La**tent **d**ynamics model, in which the latent representations evolve with an ODE enhanced by the change of observed signal and seasonality-trend characterization. We incorporate the local change of input signal into the latent dynamics in an attention-based manner and design a residual architecture over basis expansion to depict the periodicity in the underlying dynamics. To accommodate the multivariate time series forecasting, we extend the Neural Lad by learning an adaptive relationship between multiple time series. Experiments demonstrate that our model can achieve better or comparable performance against existing neural ODE families and transformer variants in various datasets. Remarkably, the empirical superiority of Neural Lad is consistent across short and long-horizon forecasting for both univariate and multivariate irregularly sampled time series.

## 1 Introduction

Achieving accurate time series forecasting has been a long-standing challenge for decades in various applications, such as traffic flow prediction [36], weather forecasts [8], management of energy consumption [1], economic analysis [10], etc. With their remarkable representation power, deep learning-based approaches are capable of modeling the complex dynamics in time series, and thus dominate in forecasting tasks recently. These methods range from recurrent neural network (RNN [24, 28]), family of neural ordinary differential equations [3, 27, 13], Transformer variants [29, 40, 5, 19, 17, 34, 41] to basis-expansion-based models (e.g. N-Beats [23]).

Our design particularly follows the family of neural ODEs, since it is an elegant as well as powerful framework to learn the temporal dynamics for time series modeling. Unfortunately, existing variants of neural ODEs, e.g. vanilla neural ODE [3], ODE-RNN [27], neural CDEs [13, 22] do not sufficiently characterize *the local change of observed signal* and ignore *inherent seasonality-trend attributes* in time series forecasting tasks. To this end, we introduce a new member to the Neural ODE family, **Neural Lad**, where the latent states' dynamics evolve with a sophisticatedly designed neural ODE. It

37th Conference on Neural Information Processing Systems (NeurIPS 2023).

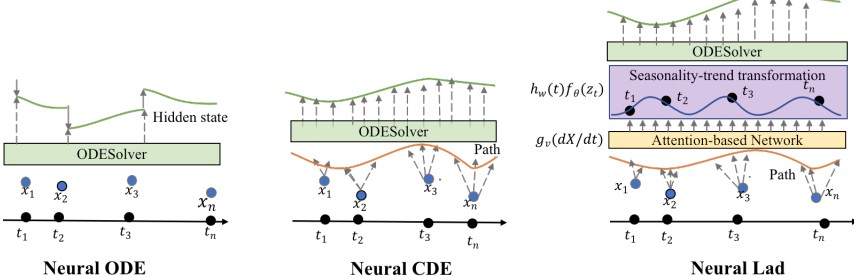

Figure 1: The sketch of Neural ODE, Neural CDE and our proposed Neural Lad for constructing hidden states from observations. **Latent ODE**: simply modifying hidden states at each observed data point; **Neural CDE**: taking the signal path into consideration which makes the hidden state continues; **Neural Lad**: enhancing the Neural CDE with transformation over input change of signal and time-dependent dynamics.

has the following characteristics tailored for forecasting challenging univariate and multivariate time series.

- We incorporate the change of interpolated observations into the latent ODE in an attention-based design to learn more flexible and expressive hidden states.
- The latent dynamics function is time-dependent, and formulated as a decomposable form w.r.t. the hidden state, the input signal and time.
- To accommodate the multivariate time series forecasting, we extend the Neural Lad through learning an adaptive relationship between multiple time series.

The comparison between our proposed Neural Lad and other related members of Neural ODEs is described in Figure. 1. The main difference is regarding how to construct the dynamics for the hidden states. Compared with Neural CDE, we employ a memory-based neural network to model how the local change of the interpolated signal path influences the latent dynamics. Further, the latent dynamic function is weighted with time-dependent seasonality-trend transformation particularly tailored for complex time series. After obtained the enhanced latent dynamics, the same as other variants of Neural ODEs, an ODESolver is used to obtain the integral hidden states. For multivariate time-series forecasting, we additionally embed an adaptive graph convolution function to encode the hidden states by incorporating the correlation with other time series (see Section 3). We then demonstrate the empirical superiority of Neural Lad across short and long-horizon forecasting for both univariate and multivariate time series data.

### Related Work

**Transformer variants for time series.** Inspired by the success of transformer-based networks in CV and NLP domains, a series of transformer variants [40, 34, 41] were proposed to capture the long term temporal dependency in time-series forecasting. Specifically, Informer [40] is an efficient network with ProbSparse attention and generative decoder. Autoformer [34] implemented an decomposition transformer network with an auto-correlation mechanism. FEDformer [41] proposed to capture the global profile of time series with seasonality-trend decomposition. However, transformer-based models consider the temporal correlation with self-attention mechanisms, where the computational complexity is high particularly when the horizon is long. Although some works like PyraFormer [21] took effort to accelerate the attention operations, the performance was degraded due to the computation-accuracy trade-off.

**Linear networks for time series.** Different from variants of transformer, works [20, 23, 37] used various types of linear blocks to extract features. N-beats [23] proposed to stack multi-layer linear blocks and employed seasonal-trend basis expansion in each layer to enhance forecasting performance, which is more time and memory efficient than transformers. SCINet [20] designed a downsample-convolve-interact architecture including multiple sample convolutions and interaction network to capture temporal correlation. Besides, [37] rethought the problem of long-horizon time-series forecasting and found that a simple one-layer fully-connected network could achieve

comparable even better prediction performance than transformer variants. However, the linear block-based networks are sensitive to data distribution and hyper-parameters according to our experiments, preventing them from generalizing to more scenarios.

**Neural ODE family.** To address the non-uniformly sampled time-series forecasting problem, neural ODE families [26, 6, 13] were introduced to model time series with continuous latent dynamics. ODE-RNN and latent ODE [26] generated interpretable hidden states through the neural ordinary differential equation. Neural CDE [13] interpolated the irregularly sampled observations into a continuous path that were then considered in latent dynamics. STG-NCDE [6] extended the Neural CDE from univariate to multivariate time-series with graph convolution to capture spatial correlation. Although Neural ODE families are more interpretable and stable than transformers and linear networks, existing Neural ODE models suffer from the ability to characterize the local change of the observed signal and the inherent periodicity-trend attributes in time series forecasting tasks.

## 2 Methodology

Let $\{x_t\}_{t=0}^T$ denote the time series, we consider time series prediction problem as

$$x_{t:t+H} = G_\Theta(x_{t-L:t}) + \epsilon, \tag{1}$$

where $H$ is the length of horizon for prediction, $L$ is the length of lookback window of historical time series, $\epsilon$ is the residual of the prediction, and $G_\Theta(\cdot)$ is parameterized neural network that will be elaborated later. Thus the loss function to learn the parameters $\Theta$ could be the mean square error or other types,

$$\min_\Theta \left\{ \mathcal{L}(\Theta) = \|x_{t:t+H} - G_\Theta(x_{t-L:t})\|_2^2 \right\} \tag{2}$$

To fully capture the intrinsic characteristics of time series data, we propose to adopt a continuous dynamical system with hidden states $z_t$ to model the entire times series, described with following neural ordinary differential equation,

$$\frac{dz_t}{dt} = F(z_{t-1}, x_{0:t-1}, t), \quad z_0 = \zeta(x_0), \quad x_t = \xi(z_t), \tag{3}$$

where $\zeta(\cdot)$ and $\xi(\cdot)$ are parameterized neural network that model the relationship $z_0 \to x_0$ and $z_t \to x_t$, respectively; $F(\cdot)$ specifies the dynamics of the hidden state and is also a neural network to be learned. We call it as **Neural Latent Dynamics** and its design is critical for modeling the time series which will be presented later.

According to the formulation provided in Eq.(3), the solution to this ODE is determined by the initial condition at $z_0$ and the latent dynamics along the integration path, i.e.

$$z_t = z_0 + \int_0^t F(z_s, x_{0:s}, s)ds, \ s \in [0, t] \tag{4}$$

This neural ODE aims to learn the hidden states at all times for time series forecasting, which can be estimated using a numerical ODE solver; and the next step prediction can be obtained by $x_t = \xi(z_t)$.

$$z_1, ..., z_t = \text{ODESolver}(F, z_0, (t_0, ..., t)) \tag{5}$$

Generally, the latent dynamics $F(\cdot)$ can be any complex form with sufficient capacity to model the sequential data. In this work, we propose a crucial design over the choice of latent dynamics function $F(\cdot)$ particularly tailored for time series, i.e. $F(z_t, x_{0:t}, t)$ is **decomposable** w.r.t. the hidden state $z_t$, the input signal $x_{0:t}$ and time $t$, named as neural decomposable latent dynamics. The decomposed $F(\cdot)$ in our paper is as:

$$F(z_t, x_{0:t}, t) = h_w(t)f_\theta(z_t)g_v(x_{0:t}), \tag{6}$$

where the vector field function $f_\theta(z_t)$ expands the hidden state $z_t$, $h_w(t)$ explicitly models the time-dependency with periodical and trend property, and $g_v$ is an attention-based network to describe how the change of observed signal influences the latent dynamics. The specification of these designs will be elaborated in Sec. 2.1 and 2.2.

We emphasize that our model choice over the latent dynamics $F(\cdot)$ is drastically different from other neural ODE family members in which only $f_\theta(z_t)$ was considered, as shown in Figure 1. The entire model can be thought as a continuous analogue of recurrent neural networks with layerwise adaptivity.

 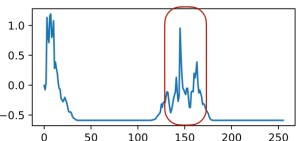 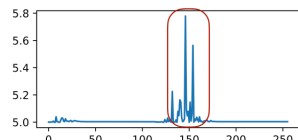

Figure 2: Visualization of attention weights. Left: attention weight; Middle: observed time series; Right: the weights of the 83rd components along time in log scale.

Here the decomposability assumption allows us to maintain a simple yet effective design over the latent dynamics without loss of expressivity, and its effectiveness will be verified empirically.

We highlight that the explicit dependence on time $t$ could both increase the expressivity and interpretability of time series modeling within the neural ODE framework, particularly for the complex time series with strong periodicity, as we experimentally illustrated in Section 4.

To achieve a more compact model, the $f_\theta(z_t, t)$ is designed as a simple multiple-layer residual neural network. In the following, we will specify two designs that aim to explicitly describe the how $g_v(x_{0:t})$ and $h_w(t)$ depends on the input signal $x_{0:t}$ and time $t$, respectively.

## 2.1 Attention-based network for modeling the change of input signal

The change of input signal $x$ can directly influence the local dynamics of the hidden states. To incorporate the input signal into the latent dynamics $f$ in a differentiable manner, we have to rely on a continuous input signal. However, the observed times series is either regular or irregular sampled sequential data $\{x_t\}_{t=0}^T$, thus, we first need to conduct a continuous approximation over the discrete time series. Inspired by the idea of neural controlled differential equation, cubic spline could be a natural alternative due to its minimum regularity for handling certain edge cases. Concretely, let $X$ denote the cubic spline with knots $t = 0, \ldots, T$ such that $X_t = (x_t, t) \in \mathbb{R}^{d_x+1}$.

With the continuous approximation $X$, typically called "signal path", the dynamic change of input signal could be depicted by the derivative $dX/dt$, and the component $g_v(x_{0:t})$ in the decomposed latent dynamics Eq (6) is specified as

$$g_v(x_{0:t}) = g_v\left(\frac{dX}{dt}\right) \in \mathbb{R}^{d_x+1}, \tag{7}$$

where $g_v : \mathbb{R}^{d_x+1} \to \mathbb{R}^{d_x+1}$ in our paper is a trainable attention-based network to describe how the change of input signal influences the latent dynamics, inspired by the idea of memory networks [30] and matching networks [31].

Concretely, let $M \in \mathbb{R}^{d_m \times (d_x+1)}$ be the memory matrix to be learned for the path gradient, $dX/dt$, shared for the entire time series. This matrix aims to memorize representative patterns of the local change along the signal path, stored as $d_m$ rows of $M$ that can be used further pattern matching. Then, for each time $t$, an attention-based transformation is employed for weighting the patterns stored in the memory matrix $M$ to obtain the final expressive control signal,

$$g_v\left(\frac{dX}{dt}\right) = M^\top softmax(M\frac{dX}{dt}). \tag{8}$$

Our memory-based modeling over the local change of the signal path is different from that of Neural CDE [13], where only the path gradient $dX/dt$ for a single time point was used. The introduction of the memory matrix allows the model to combine different patterns of local changes, which could also help to provide more discriminative features particularly for the segments with abrupt changes.

We visualize the attention weights $softmax(M\frac{dX}{dt})$ along one time series in Weather dataset in Figure 2, where memory size $d_m = 128$ and the length of time series is 250. We can observe that around $t = 150$ with obvious changes, the learned attention weights are extremely sparse and distinguishing from other segments.

## 2.2 Seasonality-trend transformation-based latent dynamics

In most of existing neural ODEs, the dynamics function only depends on $z_t$, not explicitly on the time $t$. However, in the scenario of time series, particularly for those with complex periodic and

trending characteristics, only relying on $z_t$ and $x_{0:t}$, $F(\cdot)$ might not be sufficient for capturing the hidden seasonal and trendy attributes. To this end, we propose to force the latent dynamic function to explicitly depend on the time $t$, as described by the decomposed formulation Eq (6), which forms a non-autonomous ODE in the theory of dynamical system.

With the decomposable latent dynamics, we propose to design the time-dependency function $h_w(t)$ to characterize the intrinsic seasonality and trend property based on basis expansion. It is the combination of a periodic function $h_w^{(s)}$ parameterized by $w^{(s)}$ to model the seasonal characteristics and a polynomial function $h_w^{(p)}$ parameterized by $w^{(p)}$ to model the trend features:

$$h_w^{(s)}(t) = \sum_{i=0}^{m_s} w_i^{(s)} \left( \cos(2^{i+1}\pi t) + \sin(2^{i+1}\pi t) \right) \tag{9}$$

$$h_w^{(p)}(t) = \sum_{j=0}^{m_p} w_j^{(p)} t^j \tag{10}$$

where $h_w^{(s)}(t)$ takes the form of trigonometric basis expansion with pre-specified frequencies $\{1, 1/2, 1/4, ...1/2^{m_s+1}\}$, $m_s$ and $m_p$ are the number of seasonal and polynomial basis, respectively. Then, the time-dependency factor $h_w(t)$ for the latent dynamics accounts to the sum of the two terms,

$$h_w(t) = h_w^{(s)}(t) + h_w^{(p)}(t) \tag{11}$$

Then, we multiply the vector field output $f_\theta(z_t)$ with the time-dependency function $h_w(t)$, and define it as the $l$-th basic block,

$$B(z_t^{(l)}, t) = h_w(t) f_\theta(z_t^{(l)}) \tag{12}$$

We stack multiple layers of the basic block in a residual manner that is easy to train,

$$z_t^{(0)} = z_t, \; z_t^{(1)} = z_t^{(0)} - B(z_t^{(0)}, t), \; ..., z_t^{(L)} = z_t^{(L-1)} - B(z_t^{(L-1)}, t). \tag{13}$$

Therefore, by taking the time seasonality and trend into consideration, the decomposed formulation of $F(\cdot)$ becomes,

$$F_\Theta(z_t, x_{0:t}, t) = B(z_t, t) g_v \left( \frac{dX}{dt} \right) = h_w(t) f_\theta(z_t) g_v \left( \frac{dX}{dt} \right). \tag{14}$$

where $\{\theta, w, v\}$ are parameters to be learned for the decomposed functions $h(t)$, $f(z_t, t)$ and $g(x_{0:t})$, respectively. Finally, the solution of hidden state $z_t$ can be derived as follows, which is amenable to typical neural ODE solver,

$$z_t = z_0 + \int_0^t h_w(s) f_\theta(z_s) g_v \left( \frac{dX}{ds} \right) ds, \tag{15}$$

where $f_\theta(z_t) \in \mathbb{R}^{d_z \times (d_x+1)}$ to accommodate the dimensionality of $g_v(dX/dt)$, specified as a simple residual network.

## 3 Multivariate Neural Lad

The multivariate time series forecasting typically aim to model the temporal and spatial relationship between multiple time seriese simultaneously to enhance the prediction performance. Representative models include STGCN [36], AGCRN [2] and NRI [14], among which AGCRN is the most effective and efficient approach. Inspired by AGCRN, we adaptively optimize the graph structure and learn the spatial correlation with a generalized graph convolution operation jointly.

We start with the graph convolution [15] for capturing the spatial dependency between multivariate time series for each time slice $t$,

$$\Phi_t = (I + D^{-\frac{1}{2}} A D^{-\frac{1}{2}}) Z_t \Lambda + b, \tag{16}$$

where $Z_t \in \mathbb{R}^{N \times d_z}$ represents the $t$-th slice of hidden states for the $N$ time series, and $\Phi_t \in \mathbb{R}^{N \times d_\Phi}$ collects the projected states into a matrix, $D \in \mathbb{R}^{N \times N}$ is the degree matrix, $\Lambda \in \mathbb{R}^{d_z \times d_\Phi}$ and $b$ are

the trainable weight and offset, $A \in \mathbb{R}^{N \times N}$ is the adjacency matrix, and $I + D^{-\frac{1}{2}} A D^{-\frac{1}{2}}$ is the low-order Chebyshev matrix. Therefore, Eq.(16) is the graph convolution on the hidden states to capture the correlation between multiple time series (i.e. nodes).

To make the adjacency matrix $A$ more flexible, a learnable embedding matrix $E \in \mathbb{R}^{N \times d_e}$ is designed where each row represents the embedding for each time series and $d_e$ is the dimension of node embedding. Thus, $EE^T$ could measure the similarity between these nodes. Then the normalized adjacency matrix $D^{-\frac{1}{2}} A D^{-\frac{1}{2}}$ is approximated by the following nonlinear transformation,

$$D^{-\frac{1}{2}} A D^{-\frac{1}{2}} = \phi(\sigma(EE^\top)), \tag{17}$$

where $\sigma$ is the ReLU activation function and $\phi$ is the softmax operation. We now specify a generalized version of the graph convolution in Eq. (16) to improve adaptivity and expressivity. The spatial embedding s for all the $N$ times series are defined as follows,

$$\Phi_t = \left(I + \phi(\sigma(EE^\top))\right) Z_t \Lambda \in \mathbb{R}^{N \times d_z} \tag{18}$$

where $I + \phi(\sigma(EE^\top))$ is the adaptive Chebyshev matrix inspired by AGCRN [2], and we let the dimension of the embedding of $\Phi_t$ is the same as that of the latent state $z_t$ for simplicity, i.e. $d_z$. Thus, for $i$-th time series, the spatial component corresponds to the $i$-th row of the embedding matrix $\Phi_t$

$$\phi_\gamma^{(i)}(Z_t) = \Phi_t[i, :], \tag{19}$$

where $\gamma = \{\Lambda, E\}$ denotes the collection of parameters to be learned. Then, for each time series, the hidden state $z_t$ incorporating both temporal dynamics and spatial correlation is modeled as:

$$z_t = z_0 + \int_0^t h_w(s) \phi_\gamma(Z_s) f_\theta(z_s) g_v \left(\frac{dX}{ds}\right) ds. \tag{20}$$

The Eq.(20) is the multi-variate extension of Eq.(15) with the additional spatial component $\phi_\gamma(Z_t)$.

In the encoder stage, we evaluate the time-dependent dynamics $h_w(t)$, $f_\theta(z_t)$, transformation of the signal path $g_v(dX/dt)$, then obtain the $z_t$ for all the time steps by ODESolver. In the decoder stage, we generate the hidden states and predicted outputs in a step-by-step or one-shot manner through $\xi(z_T)$.

**Discussions**    We now summarize several benefits of Neural Lad for times series modeling as follows.

1. It is the first neural ODE framework for time series in which the dynamic function explicitly depends on the hidden state $z_t$, input $x_{0:t}$ and the time $t$. It can seamlessly fit into the training procedure of neural ODE family and the model may be trained with memory-efficient adjoint-based backpropagation even across observations.

2. The explicit dependence on the time $t$ through the decomposable latent dynamics $B(z_t, t) = h_w(t) f_\theta(z_t)$ allows our neural ODE to adaptively reveal the periodicity of complex time series. Besides, the $g_v(\frac{dX}{dt})$ learns the weight of the control signal. To extend to multi-variate time series, we design the function $\psi_\gamma(z_t)$ to learn the latent dynamics from spatial correlation. This can also be verified from experimental part in Section 4.

3. When considering the relationship between input and hidden states, we employ the continuous approximation of observed discrete time series to facilitate the integration over the continuous. Thus, for the original time series either regularly or irregularly sampled, they can be fit into our model for training and inference.

## 4    Experiments

We conduct experiments on both synthetic and real-world datasets on univariate, multivariate and irregularly sampled time series forecasting tasks to demonstrate the empirical performance of our proposed approach.

Table 1: Comparison between Neural Lad and STG-NCDE on synthetic dataset.

| Horizon | STG-NCDE | | only $h_w$ | | only $g_v$ | | Neural Lad | |
|---|---|---|---|---|---|---|---|---|
| | MAE | RMSE | MAE | RMSE | MAE | RMSE | MAE | RMSE |
| 12 | 2.31 | 3.85 | 1.69 | 3.21 | 1.44 | 2.62 | **1.27** | **2.38** |
| 48 | 2.55 | 4.52 | 1.61 | 2.79 | 1.48 | 2.59 | **1.47** | **2.54** |
| 96 | 3.37 | 5.29 | 2.28 | 3.82 | 2.07 | 3.57 | **1.77** | **3.12** |

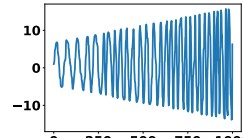 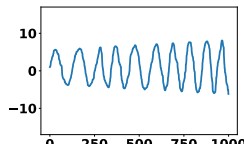 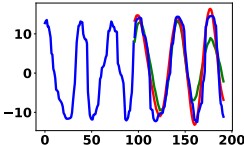 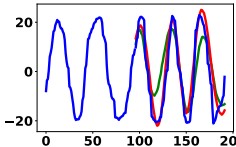

Figure 3: Left two panels: two examples of generated time series. Right two panels: prediction results. Blue: the input signal and ground truth. Green: STG-NCDE [6]. Red: Neural Lad.

## 4.1 Regularly sampled time series prediction: synthetic data

We first implement the multivariate Neural Lad on a synthetic dataset consisting of 30 periodic time series with various frequencies and linearly increasing amplitudes. The code for generating synthetic time series follows that of Neural ODE [27]. Concretely, 30 time series each with 2304 time steps are generated according to the functions $x_{i,t} = a_{i,t} \sin(2\pi b_{i,t} t + \phi) + n_{i,t}$, where $i$ is the index of each time series, $\phi$ is the phase, $n_{i,t}$ denotes the standard Gaussian noise. The amplitude and frequency ascends linearly with time $t$, i.e. $a_{i,t} = a_{i,0} + \frac{a_{max}-a_{min}}{t_{max}-t_{min}}t$ and $b_{i,t} = b_{i,0} + \frac{b_{max}-b_{min}}{t_{max}-t_{min}}t$, where for each time series the initial parameters are sampled from uniform distribution, $a_{i,0} \sim U[2, 10]$ and $b_{i,0} \sim U[0, 10]$ to increase the diversity.

Two examples of generated time series are shown in the left two panels in Figure.3. We can easily observe that these time series exhibit changing frequencies and amplitudes along time.

**Performance.** We compare a strong baseline STG-NCDE [6] with our proposed model on this synthetic dataset with prediction horizon $\{12, 48, 96\}$. The dimension of hidden states, is searched from $\{8, 16, 32, 64\}$ and we find the best setting is 16. The results are shown in Table. 1. It can be observed that Neural Lad outperforms STG-NCDE on both short and long-term forecasting. Moreover, our method improves it by $45\%$ on the short-term horizon and by $47\%$ on long-term horizon, both of which are large margins. We plot the predicted time series with horizon 90 by STG-NCDE and Neural Lad in the right two panels of Figure. 3. It can be observed that Neural Lad can fit well the overall shape of the time series.

**Ablation study.** We also conduct an ablation study by experimenting Neural Lad only with one component, periodicity $h_w(t)$ or attention-based transformation of control signal $g_v(dX/dt)$, as shown in Table 1. We find that both of the two components play important roles for enhancing performance. Moreover, the attention-based transformation $g_v$ contributes slightly more for this task.

## 4.2 Irregularly sampled time series classification: PhysioNet sepsis

One advantage of the neural ODE family, including Neural Lad, is that it naturally adapts to the case of irregularly sampled partially observed data. In this part, we apply our approach to the PhysioNet 2019 challenge on sepsis prediction [25], including 40,335 time series of variable length from the patients in ICU. Measurements are around 39-dimensional features with an hourly resolution, most of which are missing and only 10.3% of values are observed. The task is a binary classification problem to predict whether sepsis is developed during the patents' stay. We follow the experimental settings of [13], where the cases with or without observational intensity are both considered.

Since the datasets is highly imbalanced (5% positive rate), AUC is evaluated for different approaches, as reported in Table 2. We can observe that Neural Lad outperforms other alternatives when considering the observational intensity with low memory usage. Moreover, Neural Lad increases the memory less than 6% on both two classification tasks.

Table 2: Comparison between Neural Lad and other neural ODE variants on irregularly sampled dataset, PhysioNet sepsis classification.

| MODEL | TEST AUC | | MEMORY USAGE(MB) | |
|---|---|---|---|---|
| | INTENSITY | NO INTENSITY | INTENSITY | NO INTENSITY |
| GRU-ODE | 0.852 | 0.771 | 454 | 273 |
| GRU-$\Delta_t$ | 0.878 | 0.840 | 837 | 826 |
| GRU-D | 0.871 | 0.850 | 889 | 878 |
| ODE-RNN | 0.874 | 0.833 | 696 | 686 |
| NEURAL CDE | 0.880 | 0.776 | 244 | 122 |
| NEURAL LAD | **0.897** | 0.812 | 257 | 129 |

Table 3: Univariate datasets: the long horizon forecasting performance. Bold font represents the best accuracy while the underlined means the second best.

| Dataset | Horizon | ETTm1 | | ETTm2 | | ETTh2 | | Weather | |
|---|---|---|---|---|---|---|---|---|---|
| | | MSE | MAE | MSE | MAE | MSE | MAE | MSE | MAE |
| Repeat | 96 | 1.214 | 0.665 | 0.266 | 0.328 | 0.432 | 0.422 | 0.259 | 0.254 |
| | 192 | 1.261 | 0.609 | 0.340 | 0.371 | 0.534 | 0.473 | 0.309 | 0.292 |
| | 336 | 1.283 | 0.707 | 0.412 | 0.410 | 0.591 | 0.508 | 0.377 | 0.338 |
| | 720 | 1.319 | 0.729 | 0.521 | 0.465 | 0.588 | 0.517 | 0.465 | 0.394 |
| Informer [40] | 96 | 0.672 | 0.571 | 0.365 | 0.453 | 3.755 | 1.525 | 0.300 | 0.384 |
| | 192 | 0.795 | 0.669 | 0.533 | 0.563 | 5.602 | 1.931 | 0.598 | 0.544 |
| | 336 | 1.212 | 0.871 | 0.887 | 1.201 | 4.721 | 1.835 | 0.578 | 0.523 |
| | 720 | 1.166 | 0.823 | 3.379 | 1.338 | 3.647 | 1.625 | 1.059 | 0.741 |
| | avg | 0.961 | 0.734 | 1.410 | 0.810 | 4.431 | 1.729 | 0.634 | 0.548 |
| Autoformer [34] | 96 | 0.505 | 0.475 | 0.255 | 0.339 | 0.358 | 0.397 | 0.266 | 0.336 |
| | 192 | 0.553 | 0.496 | 0.281 | 0.340 | 0.456 | 0.452 | 0.307 | 0.367 |
| | 336 | 0.621 | 0.537 | 0.339 | 0.372 | 0.482 | 0.486 | 0.359 | 0.395 |
| | 720 | 0.671 | 0.561 | 0.433 | 0.432 | 0.515 | 0.511 | 0.419 | 0.428 |
| | avg | 0.558 | 0.517 | 0.327 | 0.371 | 0.450 | 0.459 | 0.338 | 0.382 |
| FEDformer [41] | 96 | 0.379 | 0.419 | 0.203 | 0.287 | 0.346 | 0.388 | 0.217 | 0.296 |
| | 192 | 0.426 | 0.441 | 0.269 | 0.328 | 0.429 | 0.439 | 0.276 | 0.336 |
| | 336 | 0.445 | 0.459 | 0.325 | 0.366 | 0.496 | 0.487 | 0.339 | 0.380 |
| | 720 | 0.543 | 0.490 | 0.421 | 0.415 | 0.463 | 0.474 | 0.403 | 0.428 |
| | avg | 0.448 | 0.452 | 0.305 | 0.349 | 0.437 | 0.449 | 0.309 | 0.360 |
| Crossformer [39] | 96 | 0.320 | 0.373 | - | - | - | - | - | - |
| | 288 | 0.404 | 0.427 | - | - | - | - | - | - |
| | 336 | - | - | - | - | - | - | 0.495 | 0.515 |
| | 720 | - | - | - | - | - | - | 0.526 | 0.542 |
| MICN  [32] | 96 | - | - | 0.179 | 0.275 | - | - | **0.161** | 0.229 |
| | 192 | - | - | 0.307 | 0.376 | - | - | 0.220 | 0.281 |
| | 336 | - | - | 0.325 | 0.388 | - | - | 0.278 | 0.331 |
| | 720 | - | - | 0.502 | 0.490 | - | - | **0.311** | 0.356 |
| | avg | | | 0.328 | 0.382 | | | 0.243 | 0.299 |
| STG-NCDE | 96 | 0.479 | 0.419 | 0.190 | 0.270 | 0.328 | 0.375 | 0.204 | 0.238 |
| | 192 | 0.384 | 0.387 | 0.259 | 0.314 | 0.430 | 0.445 | 0.263 | 0.311 |
| | 336 | 0.420 | 0.413 | 0.303 | 0.339 | 0.451 | 0.458 | 0.273 | 0.312 |
| | 720 | 0.492 | 0.461 | 2.115 | 1.029 | 0.994 | 0.689 | 0.356 | 0.347 |
| | avg | 0.444 | 0.420 | 0.717 | 0.488 | 0.551 | 0.492 | 0.274 | 0.302 |
| DLinear  [37] | avg | 0.403 | 0.407 | 0.350 | 0.401 | 0.559 | 0.515 | 0.265 | 0.317 |
| LightTS  [38] | avg | 0.435 | 0.437 | 0.409 | 0.436 | 0.602 | 0.543 | 0.261 | 0.312 |
| TimesNet  [33] | avg | 0.400 | 0.406 | 0.291 | 0.333 | 0.414 | 0.427 | 0.259 | 0.287 |
| Neural Lad | 96 | 0.337 | **0.359** | **0.172** | **0.258** | **0.275** | **0.326** | 0.162 | **0.221** |
| | 192 | **0.356** | **0.373** | **0.242** | **0.305** | **0.367** | **0.391** | **0.214** | **0.268** |
| | 336 | **0.396** | **0.406** | **0.282** | **0.333** | **0.410** | **0.422** | **0.265** | **0.303** |
| | 720 | **0.462** | **0.448** | **0.404** | **0.400** | **0.418** | **0.446** | 0.323 | **0.345** |
| | avg | **0.388** | **0.396** | **0.275** | **0.324** | **0.368** | **0.396** | **0.241** | **0.284** |

## 4.3   Real-world univaraite and mutlivariate time series prediction

**Baseline methods.**   For long-horizon univariate time series forecasting, we compare Neural Lad with SCINet, Neural CDE, DLinear [37], LightTS [38], TimesNet [33], MICN [32], Transformer variants

Table 4: The multivariate forecasting performance of Neural Lad on *PEMS* datasets.

| Model | PEMSD3 | | PEMSD4 | | PEMSD7 | | PEMSD8 | |
|---|---|---|---|---|---|---|---|---|
| | MAE | RMSE | MAE | RMSE | MAE | RMSE | MAE | RMSE |
| GraphWaveNet | 19.12 | 32.77 | 24.89 | 38.66 | 26.39 | 41.50 | 18.28 | 30.05 |
| MSTGCN | 19.54 | 31.93 | 23.96 | 37.21 | 29.00 | 43.73 | 19.00 | 29.15 |
| DCRNN | 17.99 | 30.31 | 21.22 | 33.44 | 25.22 | 38.61 | 16.82 | 26.36 |
| STGCN | 17.55 | 30.42 | 21.16 | 34.89 | 25.33 | 39.34 | 17.50 | 27.09 |
| ASTGCN | 17.34 | 29.56 | 22.93 | 35.22 | 24.01 | 37.87 | 18.25 | 28.06 |
| AGCRN | 15.98 | 28.25 | 19.83 | 32.26 | 22.37 | 36.55 | 15.95 | 25.22 |
| ST-GODE | 16.50 | 27.84 | 20.84 | 32.82 | 22.59 | 37.54 | 16.81 | 25.97 |
| Z-GCNETs | 16.64 | 28.15 | 19.50 | 31.61 | 21.77 | 35.17 | 15.76 | 25.11 |
| ST-WA | 15.17 | 26.63 | 19.06 | 31.02 | 20.74 | 34.05 | 15.41 | 24.62 |
| DSTAGNN | 15.57 | 27.21 | 19.30 | 31.46 | 21.42 | 34.51 | 15.67 | 24.77 |
| SCI-Net | 14.98 | 24.08 | 19.27 | 31.27 | 21.19 | 34.03 | 15.72 | 24.76 |
| STG-NCDE | 15.57 | 27.09 | 19.21 | 31.09 | 20.53 | 33.84 | 15.45 | 24.81 |
| Neural Lad (Ours) | **14.67** | **24.59** | **18.98** | **30.94** | **20.21** | **33.78** | **15.28** | **24.66** |

including Informer [40], Autoformer [34], Fedformer [41], Crossformer [39]. For multivariate time series considering spatial correlation, we compare our method against other graph-based deep learning approaches, including STGCN [36], ASTGCN [11], MSTGCN [12], DCRNN [18], Graph Wavenet [35], AGCRN [2], ST-WA [7], DSTAGNN [16], Z-GCNETs [4], SCI-Net [20], ST-GODE [9] and STG-NCDE [6].

**Univariate time series.** Table 3 shows the performance comparison on long-term time-series forecasting. For fair competition, we only consider the temporal latent dynamics ignoring the spatial correlation in this experiment. Specifically, we generate the hidden state with Eq.(15) instead of Eq.(20). We can observe that our proposed method outperforms all the transformer baselines by a large margin. For the model FEDformer that performs better than all other transformer variants (Autoformer, Informer, and Pyraformer), Neural Lad outperforms it on all datasets. For two linear baselines, we improved the TimesNet [33] and DLinear [37]on all dataset with the average of different horizons. The paper Crossformer [39] conduct experiments on different horizons [24, 48, 168], therefore we take it as "-" if there are no results published in the origin paper. In the same way, for dataset (*ETTm2* and *ETTh2*) without performance published in paper MICN [32], we fill it with "-". From the table 3, we can find that although Crossformer performs better at the short horizon, it fails on long horizon prediction. For TimesNet [33], Neural Lad outperform it on average of four horizons on all dataset.

**Multivariate time series.** Table 4 shows that the proposed Neural Lad achieves state-of-the-art performance on the four *PEMS* datasets on MAE metric, and also outperforms most baselines on MAPE and MSE metrics. For the baseline SCINet, although the proposed Neural Lad slightly reduces prediction error than which on PEMSD3 and PEMSD4 dataset, our method reduces MAE by 4.6% on PEMSD7 dataset and by 2.8% on PEMSD8 dataset. For neural ODE family baselines, even though STG-NCDE performs better than ST-GODE by a large margin, the proposed method improves STG-NCDE by 5.8%, 1.2%, 1.6%, 1.1% on the four PEMS datasets, respectively. Besides, our method performs slightly better than ST-WA. Overall, Neural Lad achieves the best performance for multivariate time-series forecasting in Neural ODE family.

## 5  Conclusion

In this work, we have introduced a new variant to Neural ODE family, Neural Lad, particularly tailored for accurate time series modeling and forecasting. We conduct extensive experiments on both univariate, multivariate and irregularly sampled time series datasets, Neural Lad has achieved remarkable forecasting performance in all these scenarios. Extending materials about experiment settings, computation cost, and visualizations are presented in Appendix.

In the future, we plan to improve the training efficiency of Neural Lad when considering long look-back window. This might tie with how to conduct a faster autonomous ODE solver. Another direction is to propose more architecture designs for the involved neural networks in latent dynamics model.

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
