## A Implementation details

The diagram of our proposed Neural Lad framework is illustrated in Fig.1.

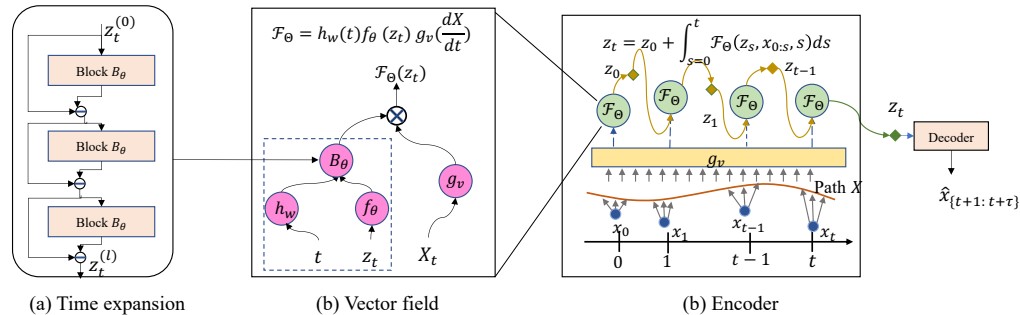

(a) Time expansion       (b) Vector field       (b) Encoder

Figure 1: Diagram of Neural Lad. **(a)** The residual architecture of time expansion for the latent dynamics; **(b)** The decomposable vector field $F(\cdot)$ w.r.t explicit time-dependent dynamics $h_w(t)$, state-dependent $f_\theta(z_t)$, local change of the interpolated signal path $g_v(\frac{dX}{dt})$; **(c)** The neural latent ODE encoder.

**The time expansion part**: we stack multiple linear layers with temporal expansion parts with residual architecture. Specifically, we add the time function $h_w$ after each linear layer, and take the state-dependent layer $f_\theta(z_t)$ and $h_w(t)$ as a basic block, then we take the residual as the input to the next layer to make the model fit the details better.

**The vector field part**: The vector field of Neural Lad consists of three parts: (i) the state-dependent layer $f_\theta$; (2) the time-dependent layer $h_w(t)$; (3) the memory network-based transformation of signal path $g_v$. In addition, we use $\ell_1$ regularization over the basis coefficients of $h_w(t)$ to avoid model sensitivity to settings of the number of basis.

**The encoder part**: Similar to Neural CDE, we solve the latent ODE with defined $F_\Theta$;

**The decoder part**: The decoder is a one-step convolution layer to generate the outputs with length $\tau$.

The pseudo code of the proposed Neural Lad is described in Alg. 1. For mult-variate time series forecasting, we multiply the function $\psi_\gamma$ to model the spatial correlations.

---

**Algorithm 1** Neural Lad for time series forecasting

---

1: **Input:** data $x_{<t}$, $t_0, ..., t_{end}$, $f_\theta$, $g_v$, $h_w$, $\psi_\gamma$, $\xi_\theta$.
2: **Output:** $\hat{x}_t, ..., \hat{x}_{t+\tau}$.
3: Initialize hidden state $z_0$, $t_0$.
4: **for** $i = 0$ **to** $t$ **do**
5:      **for** $l = 0$ **to** $L$ **do**
6:          $\hat{z}_t^{(l)} \leftarrow h_w(t) f_\theta(\tilde{z}_t^{(l-1)})$
7:          $\tilde{z}_t^{(l)} \leftarrow \tilde{z}_t^{(l-1)} - \hat{z}_t^{(l)}$
8:      **end for**
9:      **if** Multi-variate time series **then**
10:          $z_t \leftarrow \tilde{z}_t \psi_\gamma(z_t) \times g_v(dX/dt)$
11:      **else**
12:          $z_t \leftarrow \tilde{z}_t g_v(dX/dt)$
13:      **end if**
14: **end for**
15: $z_t \leftarrow ODESolver(f_\theta, g_v, h_w, \psi_\gamma, t_0, z_0)$.
16: $\{\hat{x}_{t+1}, ..., \hat{x}_{t+\tau}\} \leftarrow \xi_\theta(z_t)$

---

14

## B    Experiments details

We show the detailed settings of hyper-parameters including learning rate, hidden dimensions, weight decay, number of season frequencies, number of trends. We run all experiments on a Tesla a100-80g GPU. The training time of Neural Lad for the toy dataset is about 8s per epoch. For large real-world traffic datasets, the training time is 2-3 minutes per epoch. It is worth note that the NeuralLad converges faster than STG-NCDE, so it achieves better performance earlier than baselines.

It is worth noting that we use larger weight decay for PhysioNe sepsis dataset to avoid over-fitting. Because we exploit function $g_v$ on control gradient which increases the model fitting ability, so we need a larger weight decay and larger lr simultaneously.

Moreover, the number of seasons with different frequencies is setting to $16 - 32$ which is large than backbones, because we use l1 function which makes the selection of season frequencies and trends more flexible.

Table 1: Detailed hyper-parameter settings of all networks on all datasets.

| Tasks | Dataset | Horizons | lags | # Season | # Trend | lr | dimensions | wdecay |
|---|---|---|---|---|---|---|---|---|
| Uni-variate | ETTM1 | 96 | 48 | 16 | 4 | 0.001 | [32, 64] | 1e-3 |
| | | 192 | 96 | 16 | 4 | 0.001 | [32, 64] | 1e-3 |
| | | 336 | 192 | 16 | 4 | 0.001 | [32, 64] | 1e-3 |
| | | 720 | 192 | 16 | 4 | 0.001 | [32, 64] | 1e-3 |
| | ETTM2 | 96 | 48 | 16 | 4 | 0.001 | [32, 64] | 1e-3 |
| | | 192 | 96 | 16 | 4 | 0.001 | [32, 64] | 1e-3 |
| | | 336 | 192 | 16 | 4 | 0.001 | [32, 64] | 1e-3 |
| | | 720 | 192 | 16 | 4 | 0.001 | [32, 64] | 1e-3 |
| | ETTh2 | 96 | 48 | 16 | 4 | 0.001 | [32, 64] | 1e-3 |
| | | 192 | 96 | 16 | 4 | 0.001 | [32, 64] | 1e-3 |
| | | 336 | 192 | 16 | 4 | 0.001 | [32, 64] | 1e-3 |
| | | 720 | 192 | 16 | 4 | 0.001 | [32, 64] | 1e-3 |
| | Weather | 96 | 48 | 16 | 4 | 0.001 | [32, 64] | 1e-3 |
| | | 192 | 96 | 16 | 4 | 0.001 | [32, 64] | 1e-3 |
| | | 336 | 192 | 16 | 4 | 0.001 | [32, 64] | 1e-3 |
| | | 720 | 192 | 16 | 4 | 0.001 | [32, 64] | 1e-3 |
| Multi-variate | PEMSD3 | 12 | 12 | 16 | 4 | 0.001 | [64, 128] | 1e-3 |
| | PEMSD4 | 12 | 12 | 16 | 4 | 0.001 | [64, 128] | 1e-3 |
| | PEMSD7 | 12 | 12 | 16 | 4 | 0.001 | [64, 128] | 1e-3 |
| | PEMSD8 | 12 | 12 | 16 | 4 | 0.001 | [64, 128] | 1e-3 |
| PhysioNe Sepsis | OI | - | - | 32 | 2 | 0.0002 | [49, 49] | 0.025 |
| | No OI | - | - | 32 | 2 | 0.0002 | [49, 49] | 0.02 |

## C    Visualization

**Visualization of memory network enhanced scores.** We show the memory network enhanced control gradient comparisons in Fig.2. The middle column is the observed time series, and the right column is he weights of the 83rd components along time in log scale. We can observe that the proposed memory-enhanced control gradients can capture the oscillation of time series.

**Visualization of basis-expanded hidden states.** we visualize the hidden states of time series before and after basis expansion in Fig.4. The left column shows the hidden states before expansion, and the left is the hidden state after seasonality-trend transformation. More specifically, we first transform the hidden state with different frequencies represented with coupled sine and cosine periodical functions, then added by the polynomial functions. We can see that the hidden states after time-dependent expansion can capture more details locally, so the output network weights of Neural Lad in Fig.5 are more sparse than Neural CDE without no basis-expansion components.

**Visualization of predicted time series on short and long horizons.** we visualize the prediction performance in the test dataset in Fig.3. The left two columns are the comparison of Neural CDE and Neural Lad at short horizon with 12 steps, and the right two columns show that with horizon as 48. We can observe that our model performs significantly better at both short and long horizons. In

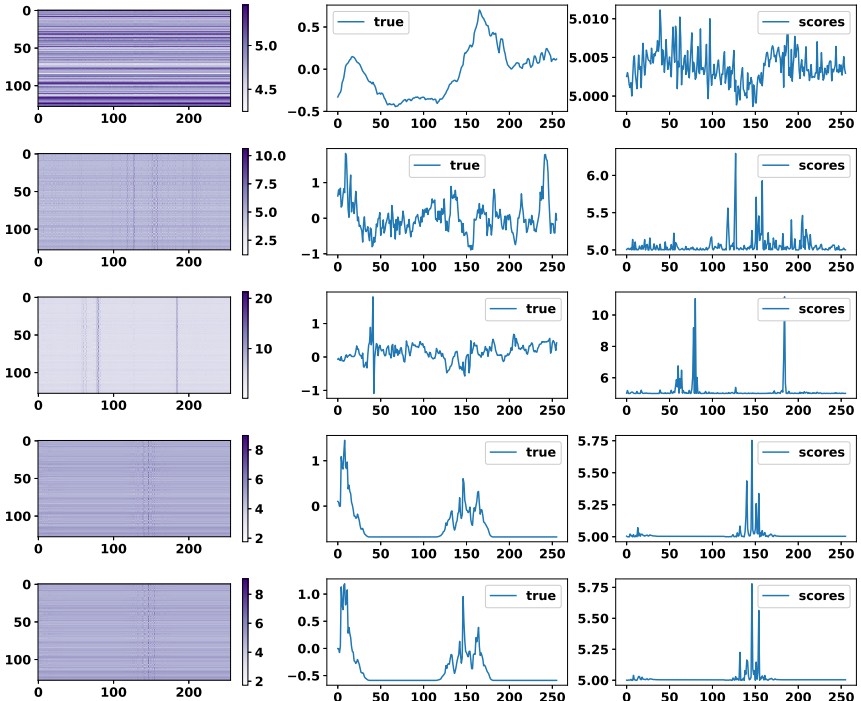

Figure 2: Visualization of memory network enhanced control gradient scores on weather dataset. Left: attention weights of memory network. Middle: the observed time series. Right: the weights of the 83rd components along time in log scale.

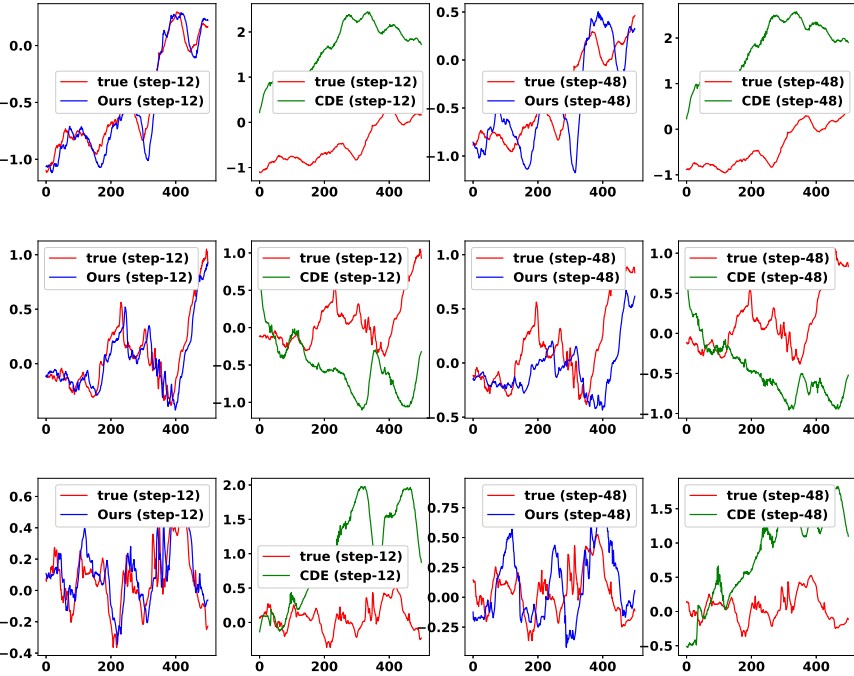

Figure 3: Visualization of predicting results comparisons between Neural CDE and Neural Lad (ours). The left two figures are performances at step-12 (short horizon) and the right two figures are results at step-48 (long horizon)

.

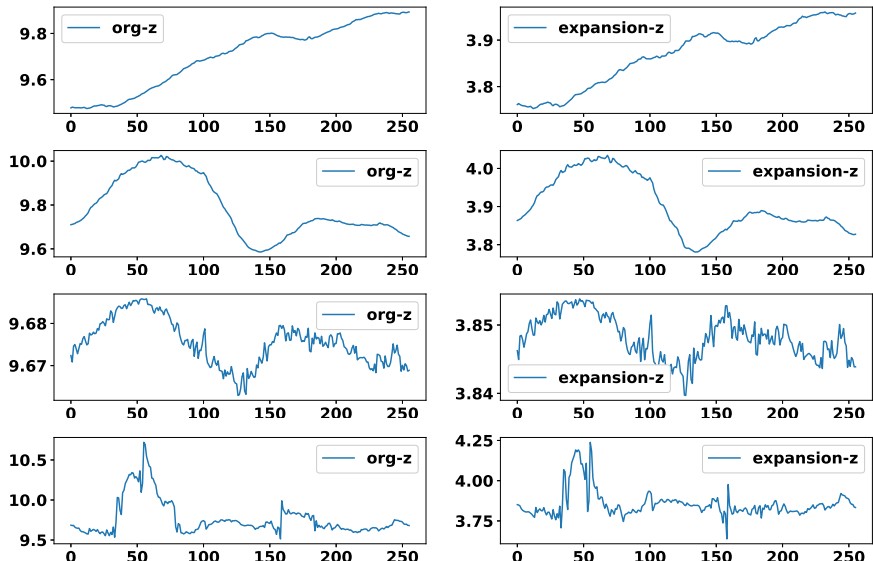

Figure 4: Visualization of basis expansion transformation on weather dataset. Left figure: hidden states before basis expansion. Right figure: hidden states after basis expansion including season and trend transformation.

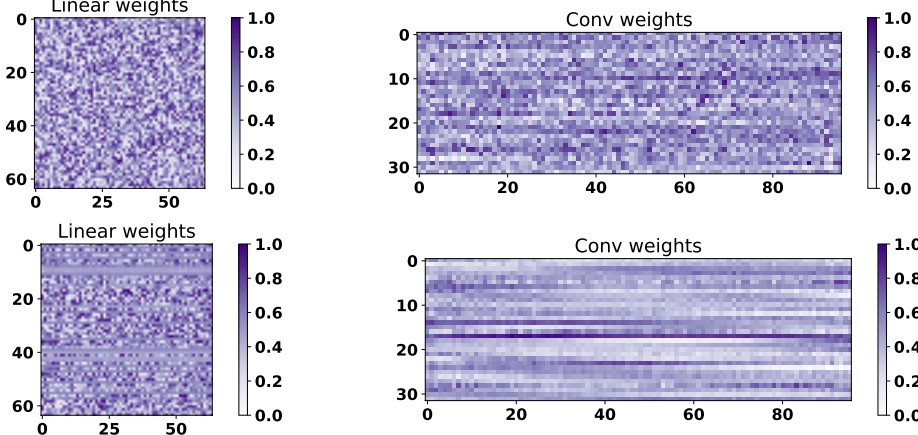

Figure 5: Visualization of network weights. The above two figures are the output linear weight and end convolution weights of Neural Cde. The bottom two figures are the linear weights and end convolutions weights of Neural Lad.

43  addition, both models fit well on short-distance prediction while it is hard to predict long horizons
44  accurately.

45  **Visualization of optimized network weights.** we visualize the learned network weights including
46  the linear weights of the vector field function $f_\theta$ and the end-convolution weights of function $\xi_\theta$ for
47  Neural CDE and our proposed Neural Lad. For a fair comparison, we normalize the model parameters
48  of two layers to the same range. We can observe that the network weights of Neural Lad are more
49  sparse than those of Neural CDE, demonstrating that the proposed two components of Neural Lad
50  can capture the hidden dynamic better so it is not necessary to fit the future time series with more
51  parameters.