# OpenReview forum: "Neural Lad: A Neural Latent Dynamics Framework for Times Series Modeling"
_NeurIPS.cc/2023/Conference — NeurIPS 2023 poster_

### Official Review · Reviewer_85MJ · 2023-06-24

**Soundness:** 3 good
**Presentation:** 3 good
**Contribution:** 3 good
**Rating:** 6
**Confidence:** 3

**Summary:**

The authors propose a new neural ODE framework, Neural Lad (Neural Latent dynamics model), that decomposes the differential function in three components: a NN differential function, an attention-based network,  a time-dependency function and a graph convolution network for spatial correlations. They evaluate the method on short and long-horizon forecasting tasks for both univariate and multivariate time series data. The proposed method outperforms or performs on par with the existing Neural ODE, time-series transformer, and graph NN based models.




**Strengths:**

Originality:
The work is a novel combination of known techniques and it is clear how it compares to existing work, for example,  Neural CDEs.
The related work section is well organized and all the related methods have been addressed, providing an adequate overview of the related work for the reader.

Clarity:
The submission is very clearly written. Therefore it is easy to follow the proposed method.


Significance:
The authors provide a novel NODE based method for long-horizon forecasting



**Weaknesses:**

Quality:

The authors introduce a function $h_w(t)$ which should extract seasonality trend. However, neither mathematically nor experimentally, it isn't clear whether this function really achieves this. Traditionally trend is extracted as the mean value of the data, while seasonal part would be the remaining time series data, when trend is removed. From the introduced equations, where the output of the differential function is scaled by the time-dependency factor is at the moment unclear how this is achieved.

Similarly, the authors introduce an attention-based network to model change of the input signal, but the obtained weight is multiplied with the 'memory matrix' rather than the input signal itself, thus it is unclear, what information does this memory matrix has learned that reflects the input signal. This matrix is also not visualized.

Clarity:

For eq. (1) and eq (2) I would recommend the authors to update the prediction to $\hat{x_{t:t+H}}$. So then in eq. (2) L2 loss is between the true observations and the predictions by the network.

Line 92, the relationship for the fist function $\xi$ seems to be in the wrong direction, given whats defined in eq. (3).

For the experiments section there are crucial points missing: how many datapoints used as input, dataset details, etc. This information should be provided in the supplementary material.

Under the hyperparameter section there are also missing details on what type of solver was used, solver parameters, what type of optimizer, etc.

Section 2.2. the block residual architecture is unclear from the text, I would recommend moving the figure from the appendix to main text.

Significance:
The results show that the proposed method indeed achieves an improved performance compared to the existing models. But I would like that the authors would have provided some further analysis, results on the different decomposition components, to strengthen their claims of the purpose of the different functions. Moreover, from a theoretical standpoint it is unclear under which conditions $f_\theta(z_t)$ still remains the correct underlying differential equation.

**Questions:**

Eq.6 The differential equation F() is decomposed in 3 components, differential function, attention-network, time-dependency network. This function, F() is used in the ODESolver that iteratively class this function for dt increments. For every increment, the model goes through some number of blocks (L), where the output of the differential function is scaled by the time-dependency function. Could you please clarify what is the output of $h_w(t)$, and why would you scale the state-dependent dynamics by it?

Do you have some additional results where it can be seen that this multiplication identifies seasonal trends as seen in the data?

Is the extrapolation (forecasting) autoregressive or you have to pre-define the output sequence length?

For Eq.8 from self-attention perspective we have keys, queries in values. In relation to your work, I would identify the values as $\frac{dX}{dt}$ rather than M. Why do you apply the softmax weights to the memory matrix rather than the signal path, which you subsequently use to update the vector from the differential equation?


What is the number of parameters, complexity, compute time of the proposed method? As cubic spline is an approximation method as well as the time expansion results in additional residual connections, how does it compare to existing NODE models and the transformer architectures?

Experiments.
For all experiments, in the final version, I would like to ask the authors to also report the std across multiple runs of the model.

It is unclear how many timepoints were taken as input to for example to compute the attention-based network output?

For simulated data, are there more visualization that showcase that the model has captured the linear trend in the data?
Similarly, for the ablations on the simulated data when the $h_w$ component is removed, does the model then fail to capture the linear trend?
Is there a clear pattern that the $g_v$ function is beneficial for time-series that have sudden changes, like weather forecasting?

As the authors have changed the RHS of the differential equation, is there a theoretical proof that $f_\theta(z_t)$ still learns the correct underlying dynamics? What are the necessary/sufficient conditions under which this is the case?





**Limitations:**

The authors have briefly touched upon the training efficiency of Neural Lad, however, I would recommend the authors to include a more thorough discussion on this, especially as  in the related work section this is mentioned as the major draw back for transformer based models, however, it is not explicitly discussed for the model at hand. This is information is also missing in the supplementary material.

---

> ### Author Rebuttal · Authors · 2023-08-09
>
> We thank the reviewer's recognition and valuable comments on the contribution of our work. We address the concerns in the following.
>
> **Weakness Part:**
>
> **Quality:**
>
> **Q:** The mechanism of time-dependent component
>
> **R**: The effectiveness of the designed time-dependent function $h_w(t)$ could be verified from various aspects.
>
> **i)** Empirically, according to our ablation study, Table.1 shows that using $h_w$ for extracting seasonality and trend  could improve the prediction performance.
>
> **ii)** From the perspective of underlying mechanism, typically seasonality is  a regular, cyclical, recurring fluctuation. Therefore,  we propose to constrain the seasonality component of $h_w(t)$ to belong to the class of periodic functions, and the natural choice for the basis is the Fourier series. The trend is that most of the time it is  a slowly varying function. In order to mimic this behaviour we propose to constrain to be a polynomial of small degrees, a function slowly varying across forecast window. Our modeling over seasonality and trend is reasonable and different from the traditional one  using mean of time series and its residual.
>
> **iii)** With this careful design, the learned latent dynamics of $z_t$ could learn more sophisticated details than that without such design, as demonstrated in Figure 4 in Appendix.
>
>
> **Q:** The mechanism of attention-based network
>
> **Reply:** Our attention-based modeling over the change of the observed signal is inspired by memory networks [30] and matching networks [31]. The memory matrix $M$ should be learned,  and can be seen as a memory bank, where each row represents a particular pattern describing one aspect of input signal's local change.  The introduction of  the memory matrix allows the model to combine different patterns of local changes, which could also help to provide more discriminative features particularly for the segments with abrupt changes. In Fig. 2 of Appendix, we visualize the memory weight and the input signals, we can observe the learned weights are obviously distinguished from others at sudden changed points.
>
>
> **Clarity:**
>
> **Q:** clarity of eq. (1) and (2)
>
> **Reply:** We would follow your advice for the notation, where actually the prediction $\hat{x}_{t:t+H} = G_{\Theta} (x_{t-L:t})$.
>
> **Q:** the wrong direction in eq. (3).
>
> **Reply:**  We apologize for the typo here.  The last equation of (3) should be $x_{t:t+H} = \xi(z_t)$, denoting the decoding process. We use the learned dynamics $z_t$ to predict the following $H$ steps, as illustrated by Figure 1 in Appendix.
>
>
> **Q:**  Significance of the proposed components
>
> **Reply:** From the performance point of view, we present an ablation study in Table 1, from which we can observe both the time-dependent $h_w(t)$ and memory-based component for input signals improve performance.
> Also according to the visualization, we can see that the learned weights are more sparse than Neural CDE in Fig. 5, indicating Neural Lad tends to avoid overfitting.
> From a theoretical standpoint, we think the time-dependent component $h_w(t)$  makes the model learn the weights of basis expansion instead of direct fitting, therefore the network weights are sparse than neural CDE.
> Moreover, CDE only deals with the case when the hidden dynamic is linear to the change of input signal, which may fail when the control signal is non-linear.  Therefore we propose an attention-based network to model the local change of observation signals.
>
> **Questions Part**:
>
> **Q:**
> "...Could you please clarify what is the output of $h_w(t)$, and why would you scale the state-dependent dynamics by it?"
>
> **Reply:**  The output of  $h_w(t)$  is a scalar, indicating the instant strength of periodicity and trend. This scaling is one modeling choice of incorporating the periodicity and trend into the dynamics of latent z.
>
> **Q:**
> "Do you have some additional results where it can be seen that this multiplication identifies seasonal trends as seen in the data?"
>
> **Reply:** Empirically, according to our ablation study for synthetic datasets with very strong peoridicity and trend property, the experiment results in the Table.1 shows that using $h_w$  could improve the prediction performance. Additionally, with the employment of this scaling, the learned latent state $z$ shows more fine-grained details, as plotted in Fig. 4 of Appendix.
>
>
> **Q:**
> Running time of Neural Lad.
>
> **Reply:**  We add the complexity analysis and computational time comparison with other approaches, as shown in Figure 1 in the attached pdf file.
> We can observe that the NeuralLad converges faster than STG-NCDE, so it achieves better performance earlier than baselines.
>
> For the prediction accuracy comparison with existing neural ODE and transformer models, we show them in Table 1-4 for different tasks in the main paper.
>
> **Q:**  "Experiments. For all experiments, in the final version, I would like to ask the authors to also report the std across multiple runs of the model."
>
> **Reply:** For the current version, as we fix the random seed the same as the baseline models, the experiment results are fixed when running multiple times for both uni-variates and multi-variate time-series forecasting tasks. Indeed, we will also report the std across multiple runs for different random seeds.
>
> **Q:**  Visualization of toy data
>
> **Reply:** In the right two panels of Figure 3 in main paper, we show that without consideration of trend and seasonality, the neural CDE model underestimates the rising trend near the peak and thus cannot capture the true waveform accurately.
>
> **Q:**
> whether neural lad can learn the correct dynamics?
>
> **Reply:** It is straightforward to show that our model can still learn the correct underlying dynamics, since our Neural Lad is a non-trivial generalization of original neural CDE that has already been proved to be able to sufficiently expressive.
>
> The details on the datasets and hyperparameter settings are described in Appendix

---

> > ### Comment · Reviewer_85MJ · 2023-08-12
> >
> > Thank you for the detailed response and clarifications on my questions. I will raise my score.
> >
> > In the final version I would still like to see the model performance across different initialization seeds. And I would still like to see the following ablation: a visual result of the generated time series by Neural Lad for Table 1. with (a) only $h_w$ and (b) only $g_v$. As one would assume that in setting (b) the model with not capture periodical/trend property; while in (a) the latent dynamics should be less fine grained as the observed signal does not affect the latent dynamics. (Correct me if these assumptions are wrong).

---

> > > ### Author Response · Authors · 2023-08-14
> > > **More experiments and visualizations**
> > >
> > > We appreciated your recognition on our clarifications and further suggestions that could definitely make a better version of our work. As you suggested, we will add more descriptions on the statistical results on the model performance and visualizations to show our model can learn more details than that without consideration of periodicity/trend property.

---

### Official Review · Reviewer_GHEt · 2023-07-04

**Soundness:** 3 good
**Presentation:** 3 good
**Contribution:** 3 good
**Rating:** 6
**Confidence:** 3

**Summary:**

This paper addresses the problem of characterizing the local change of observed signals and ignoring inherent periodical property in time series forecasting tasks. A new neural ODE-based framework is proposed with 1) a decomposable latent space for time-dependent dynamics and 2) an attention-based design for local changes in observation. The framework is further extended to multivariate settings using graph-based networks to adaptively learn the spatial correlation. Experiments have been presented on both univariate and multivariate settings to demonstrate the improved forecasting performance of the proposed model.

**Strengths:**

1. The presentation of the dynamic model is clear and well-structured. The attention-based networks handle the local change of the signal and the time-dependency function characterizes the seasonal and trend properties of the signal.
2. Empirical results show improved forecasting performance of the proposed model on both synthetic and real-world datasets.

**Weaknesses:**

1. I feel that the approach is somewhat incremental from the perspective of the methodology in that it is an extension of the previous works (e.g. Kidger et al 2020, Choi et al 2022), in combination with a decomposable form of latent dynamics and attention-based feature extractor. Could the author elaborate more on how the proposed method differs from the previous works?
2. Experimental settings could be more elaborated: 1) The motivation of choosing baseline models in Table 2, Table 3, and Table 4, and how the proposed model improved could be more detailed; 2) The author should also provide more information about datasets used in the two univariate and multivariate settings in terms of their local property or seasonal features. I also think the ablation study is not sufficient: 1) There should be evidence to prove the benefits of using attention-based networks for local changes; 2) The detail of improvements in spatial relationships should also be provided.


**Questions:**

Please check the weaknesses mentioned above.

**Limitations:**

N.A.

---

> ### Author Rebuttal · Authors · 2023-08-09
>
> We thank the reviewer's recognition and valuable comments on the contribution of our work. We address the concerns in the following.
>
> **Weakness Part:**
>
> **Q:** "I feel that the approach is somewhat incremental from the perspective of the methodology in that it is an extension of the previous works (e.g. Kidger et al 2020, Choi et al 2022), in combination with a decomposable form of latent dynamics and attention-based feature extractor. Could the author elaborate more on how the proposed method differs from the previous works?"
>
> **Reply:** As shown in the reply to reviewer LTR9, the Neural Lad is a new member of neural ODE family. The idea of constructing a decomposable design for latent dynamics for time series modeling is novel and effective according to our knowledge over literature. We emphasize that our model choice over the latent dynamics F(·) is drastically different from othe neural ODE family members in which only $f_θ(z_t)$ was considered, as shown in Figure 1. The entire model can be thought as a continuous analogue of recurrent neural networks with layerwise adaptivity. Here the decomposability assumption allows us to maintain a simple yet effective design over the latent dynamics without loss of expressivity, and its effectiveness has been verified empirically.
>
> Compared to Neural CDE (Kidger et al 2020,) and STG-NCDE (choi et al 2022), one contribution is to model the explicit time dependence on time $t$ with the decomposable time dynamics. In Figure 5 of Appendix, we can observe that learned linear weights and convolution weights are more sparse than Neural CDE by considering the seasonal and trend time dynamics, which indicates that the proposed component can capture the hidden dynamics better so it is not necessary to fit the future with more parameters.
> The other contribution is to use a memory network on the control gradient to capture the local change of the observed signals, which can improve the performance with a large margin (shown in the Table 1), and model the sudden change of input signals (visualized in Figure 2).
>
>
>
> **Q:** "Experimental settings could be more elaborated...."
>
> **Reply:**
> 1) The principle of selecting baselines for comprison is that we identify recently works that have competitive and even state-of-the-art prediction performance. Concretely, in Table 2, the PhysionNet sepsis classification task, we choose the same baselines as Neural CDE.
> In Table 3, for the univariate time-series forecasting, we choose three kinds of models including the widely-used transformer models (Autoformer, Fedformer), the light-weighted linear network (LightTS, DLinear) and Neural CDE network (STG-NCDE).
> In Table4, for the multi-variate time-series datasets, we choose the graph time-series model such as STGCN, AGCRN, DSTAGNN and STG-NCDE as baselines.
>
> 2) Regarding the details of used datasets for testing, we will follow your advice and  add more description to make them more self-contained in the revision. Thank you for your valuable suggestions.
>
> 3) Regarding the ablation study:from the performance perspective, we show the benefits of attention-based network in Table 1. Specifically, the $g_v$ in Table 1 is the memory network component, we can observe that the MAE degrades from $2.31$ to $1.44$ on horizon $12$ and from $3.37$ to $2.07$ on horizon $96$ by only adding $g_v$ to STG-NCDE, which demonstrates the benefits of memory network.  From the explanation perspective, we visualize the time-series and attention weight of the memory network in Figure 2 in Appendix, we can observe that the learned weights are extremely sparse and distinguish from others when the time-series has sudden changes.
>
> The benefit of considering spatial relationship has already been verified by STG-NCDE  that can improve the performance of Neural CDE with a large margin on the traffic dataset, for example, STG-NCDE improves Neural CDE from $20.44$ to $15.57$ on PEMS3, and from $26.31$ to $19.21$ on PEMS4. Neural Lad can be seen as a non-trivial extension over STG-NCDE, which also enjoys the advantage of considering spatial relationship, as shown in Table 2.

---

> > ### Comment · Reviewer_GHEt · 2023-08-16
> >
> > Thank you for your clarification and additional details about the experimental setting.

---

### Official Review · Reviewer_LTR9 · 2023-07-06

**Soundness:** 4 excellent
**Presentation:** 4 excellent
**Contribution:** 3 good
**Rating:** 6
**Confidence:** 4

**Summary:**

This paper presents a new framework for modeling time series using a controlled latent neural-ODE-based dynamics model. The proposed latent dynamics function uses a special factorized structure, which effectively disentangles the influences of time (via a periodic basis expansion to capture periodic patterns), the current latent state, and the input signal's history (leveraging an attention-based architecture). Both uni- and multivariate versions of the model are presented. Numerical validations are conducted across a variety of tasks and benchmarked against a number of neural ODE and transformer variants.

**Strengths:**

* The paper is generally well written and easy to follow.
* The numerical validation is quite extensive, spanning over tasks of different natures and (short- and long-term) prediction regimes. The proposed method is compared against a wide range of variants in the neural ODE and transformer model families.


**Weaknesses:**

* The idea itself is not the most novel, involving a simple factorization and model architectures drawing cues from popular ones already existing in the literature. However, this is largely compensated by the exhaustive numerical validations where consistent improvements are observed.
* Discussion on the computation costs and potential limitations is missing.


**Questions:**

* Can you comment on the cost aspects (i.e. training, inference and memory footprints) of the Neural Lad models? I am curious how it compares to the baselines tested in the paper.
* The notation across Equations (13-14) is slightly unclear to me. How does $B(z_t, t)$ in equation (14) relate to the $B$ operations in Equation (13)?


**Limitations:**

Not much discussion on the limitations.

---

> ### Author Rebuttal · Authors · 2023-08-09
>
> We thank the reviewer's recognition and valuable comments on the contribution of our work. We address the concerns in the following.
>
> **Weakness Part:**
>
> **Q:** "The idea itself is not the most novel, involving a simple factorization and model architectures drawing cues from popular ones already existing in the literature. However, this is largely compensated by the exhaustive numerical validations where consistent improvements are observed."
>
> **Reply:** We admit that our model is not a completely new model, since it is a new member of neural ODE family. However, **the idea of constructing a decomposable design for latent dynamics for time series modeling is novel and effective** according to our knowledge over literature. We emphasize that our model choice over the latent dynamics F(·) is drastically different from othe neural ODE family members in which only $f_θ(z_t)$  was considered, as shown in Figure 1. The entire model can be thought as a continuous analogue of recurrent neural networks with layerwise adaptivity. Here the decomposability assumption allows us to maintain a simple yet effective design over the latent dynamics without loss of expressivity, and its effectiveness has been verified empirically.
>
> Compared to Neural CDE (Kidger et al 2020,) and STG-NCDE (choi et al 2022), one contribution is to model the explicit time dependence on time $t$ with the decomposable time dynamics. In Figure 5 of Appendix, we can observe that learned linear weights and convolution weights are more sparse than Neural CDE by considering the seasonal and trend time dynamics, which indicates that the proposed component can capture the hidden dynamics better so it is not necessary to fit the future with more parameters.
> The other contribution is to use a memory network on the control gradient to capture the local change of the observed signals, which can improve the performance with a large margin (shown in the Table 1), and model the sudden change of input signals (visualized in Figure 2).
>
> **Q:** "Discussion on the computation costs and potential limitations is missing."
>
> **Reply:** We add the complexity analysis and computational time comparison with other approaches in Figure 1 in the attached pdf file. We run all experiments on a Tesla a100-80g GPU. The training time of Neural Lad for the toy dataset is about 8s per epoch, the forecasting time is 0.4s for every validation iteration (0.4/265=0.0015s).  For large real-world traffic datasets (such as PEMAS03, PEMAS04), the training time is 2-3 minutes per epoch. We visualize the training process on Toy dataset in Figure 1. We can observe that the NeuralLad converges faster than STG-NCDE, so it achieves better performance earlier than baselines.
>
>
> **Questions Part:**
>
> **Q:** Can you comment on the cost aspects (i.e. training, inference and memory footprints) of the Neural Lad models? I am curious how it compares to the baselines tested in the paper.
>
> **Reply:** Compared to its main baseline neural CDE, our model does not increase too much overhead; remarkably, NeuralLad converges faster than STG-NCDE, so it achieves better performance earlier than the baseline, as shown by  Figure 1 in the attached pdf file.
>
> **Q:** "The notation across Equations (13-14) is slightly unclear to me. How does $B(z_t, t)$ in equation (14) relate to the  $B$ operations in Equation (13)?"
>
> **Reply:**  The basis expansion component $B(z_t, t)$ in eq.(14) is a multi-layer stacked residual network as shown in eq.(13),  where the $L$ is the number of layers of the basis network. In addition, the structure of $B(z_t, t)$ is shown in Fig. 1.(a) in Appendix.

---

> > ### Comment · Reviewer_LTR9 · 2023-08-14
> >
> > Thank you for your clarifications and providing additional information on the computation costs.

---

### Official Review · Reviewer_gsAa · 2023-07-11

**Soundness:** 3 good
**Presentation:** 4 excellent
**Contribution:** 3 good
**Rating:** 7
**Confidence:** 2

**Summary:**

This paper propose a novel neural ordinary differential equation framework for time series modeling. The main contribution is in the design of latent dynamic function $F(\cdot)$ which can be decomposed into hidden state dynamics $f_\theta(z_t)$, time-dependency with periodical and trend property $h_w(t) $, and attention-based network to model signal to latent dynamics effect $g_v(x_0:t)$.
Also, they showed that the model can be extend to a multivariate time series forecasting. The proposed design is theoretically sound and authors proved them to be effective on synthetic and real data.

**Strengths:**

- The paper is well-written and the contribution is well stated.
- The proposed design is compared thoroughly with different types of baseline in various synthetic/real data and outperforms baseline in most cases.

**Weaknesses:**

- Analysis on the computational complexity/cost is missing

**Questions:**

- Synthetic data seems to have only trend dynamics. It would be nice to see the result from seasonal(periodic) dynamics

---

> ### Author Rebuttal · Authors · 2023-08-09
>
> We thank the reviewer's recognition on the novelty and contribution of our work. We address the concerns in the following.
>
>
> **Weaknesses Part**:
>
> **Q**: Analysis on the computational complexity/cost is missing
>
> **Reply**: We add the complexity analysis and computational time comparison with other approaches, as shown in Figure 1 in the attached pdf file. We run all experiments on a Tesla a100-80g GPU. The training time of Neural Lad for the toy dataset is about 8s per epoch, the forecasting time is 0.4s for every validation iteration (0.4/265=0.0015s).  For large real-world traffic datasets (such as PEMAS03, PEMAS04), the training time is 2-3 minutes per epoch. We also visualize the training process on Toy dataset in Figure 1 .
> We can observe that the NeuralLad converges faster than STG-NCDE, so it achieves  better performance much earlier than the baseline.
>
>
> **Questions Part**:
>
> **Q**: Synthetic data seems to have only trend dynamics. It would be nice to see the result from seasonal(periodic) dynamics
>
> **Reply**:
> The generative formula of the synthetic data is $x_{i,t}= \sin(2\pi
>  b_{i,t} t + \phi) + n_{i, t}$, **including both periodicity and trend dynamics**,  where the changing of frequency $b_{i,t}$ and amplitude $a_{i,t}$ represent the seasonality and trend,  respectively.

---

> > ### Comment · Reviewer_gsAa · 2023-08-17
> >
> > Thank you for providing detailed information on the computation costs.

---

### Author Rebuttal · Authors · 2023-08-09

We thank all the reviewers' recognition and valuable comments on our work. We have carefully responsed the concerns raised for each reviewers, including clarification of the novelty, adding more exprimental results. We attached the additional experimental results  as a pdf file for further check.

---

### Decision · Program_Chairs · 2023-09-21

**Decision:**

Accept (poster)

**Comment:**

This paper proposes a specific functional form for the dynamics function in a neural ODE. The proposal function has three components: a component to model the time-dependency (e.g. seasonal trends), a function to model the effect from the previous history on the dynamics (parameterized via a attention network) and a state dependent function. There is some novelty put into the intelligent design of the function h to model seasonal trends using a trignometric basis expansion with pre-specific frequencies. Overall, this is a simple idea and much of the novelty of this paper lies in the intelligent design for the neural architecture of the dynamics function. Overall the reviewers found the work clear, accessible and even if incremental at times. The empirical evaluation is very thorough. The model is carefully studied in a variety of benchmark datasets and compared to an extensive suite of related work. Overall this is a strong empirical paper and I'm inclined to accept.